# Synthesis of New Triazole-Based Thiosemicarbazone Derivatives as Anti-Alzheimer’s Disease Candidates: Evidence-Based In Vitro Study

**DOI:** 10.3390/molecules28010021

**Published:** 2022-12-20

**Authors:** Fazal Rahim, Hayat Ullah, Muhammad Taha, Rafaqat Hussain, Maliha Sarfraz, Rashid Iqbal, Naveed Iqbal, Shoaib Khan, Syed Adnan Ali Shah, Marzough Aziz Albalawi, Mahmoud A. Abdelaziz, Fatema Suliman Alatawi, Abdulrahman Alasmari, Mohamed I. Sakran, Nahla Zidan, Ibrahim Jafri, Khalid Mohammed Khan

**Affiliations:** 1Department of Chemistry, Hazara University, Mansehra 21120, Pakistan; 2Department of Chemistry, University of Okara, Okara 56130, Pakistan; 3Department of Clinical Pharmacy, Institute for Research and Medical Consultations (IRMC), Imam Abdulrahman Bin Faisal University, Dammam 31441, Saudi Arabia; 4Department of Zoology, Wildlife and Fisheries, Sub-Campus Toba Tek Singh, University of Agriculture Faisalabad, Punjab 36050, Pakistan; 5Department of Agronomy, Faculty of Agriculture and Environment, The Islamia University of Bahawalpur, Bahawalpur 63100, Pakistan; 6Department of Chemistry, University of Poonch, Rawalakot 12350, Pakistan; 7Faculty of Pharmacy, Universiti Teknologi MARA Cawangan Selangor Kampus Puncak Alam, Bandar Puncak Alam 42300, Selangor, Malaysia; 8Atta-ur-Rahman Institute for Natural Product Discovery (AuRIns), Universiti Teknologi MARA Cawangan Selangor Kampus Puncak Alam, Bandar Puncak Alam 42300, Selangor, Malaysia; 9Department of Chemistry, Alwajh College, University of Tabuk, Tabuk 47512, Saudi Arabia; 10Department of Chemistry, Faculty of Science, University of Tabuk, Tabuk 71491, Saudi Arabia; 11Department of Biochemistry, Faculty of Science, University of Tabuk, Tabuk 71491, Saudi Arabia; 12Department of Biology, Faculty of Science, University of Tabuk, Tabuk 71491, Saudi Arabia; 13Biochemistry Section, Chemistry Department, Faculty of Science, Tanta University, Tanta 31527, Egypt; 14Department of Nutrition and Food Science, Faculty of Home Economics, University of Tabuk, Tabuk 71491, Saudi Arabia; 15Department of Home Economics, Faculty of Specific Education, Kafr ElSheikh University, Kafr ElSheikh 33516, Egypt; 16Department of Biotechnology, Faculty of Sciences, Taif University, Taif 21944, Saudi Arabia; 17H.E.J. Research Institute of Chemistry, International Center for Chemical and Biological Sciences, University of Karachi, Karachi 75270, Pakistan

**Keywords:** triazole, thiosemicarbazone, acetylcholinesterase, butyrylcholinesterase, structure activity relationship, molecular docking study

## Abstract

Triazole-based thiosemicarbazone derivatives (**6a–u**) were synthesized then characterized by spectroscopic techniques, such as 1HNMR and 13CNMR and HRMS (ESI). Newly synthesized derivatives were screened in vitro for inhibitory activity against acetylcholinesterase (AChE) and butyrylcholinesterase (BuChE) enzymes. All derivatives (except **6c** and **6d**, which were found to be completely inactive) demonstrated moderate to good inhibitory effects ranging from 0.10 ± 0.050 to 12.20 ± 0.30 µM (for AChE) and 0.20 ± 0.10 to 14.10 ± 0.40 µM (for BuChE). The analogue **6i** (IC_50_ = 0.10 ± 0.050 for AChE and IC_50_ = 0.20 ± 0.050 µM for BuChE), which had di-substitutions (2-nitro, 3-hydroxy groups) at ring B and tri-substitutions (2-nitro, 4,5-dichloro groups) at ring C, and analogue **6b** (IC_50_ = 0.20 ± 0.10 µM for AChE and IC_50_ = 0.30 ± 0.10 µM for BuChE), which had di-Cl at 4,5, -NO_2_ groups at 2-position of phenyl ring B and hydroxy group at ortho-position of phenyl ring C, emerged as the most potent inhibitors of both targeted enzymes (AChE and BuChE) among the current series. A structure-activity relationship (SAR) was developed based on nature, position, number, electron donating/withdrawing effects of substitution/s on phenyl rings. Molecular docking studies were used to describe binding interactions of the most active inhibitors with active sites of AChE and BuChE.

## 1. Introduction

Alzheimer’s disease (AD) is an irreversible, neurodegenerative and progressive disorder of the brain that diminishes the cholinergic system and results in disorientation, memory loss, impaired ability to solve problems and impaired cognition [1,2,3]. AD is the major cause of dementia in aging populations. The acetylcholinesterase (AChE) and butyrylcholinesterase (BuChE) enzymes cause apoptosis of neuronal cells by plaques formed by aggregation of neurotoxic beta amyloid. They are involved in hydrolysis of acetylcholine to generate acetic acid and choline, leading to shortening of duration of acetylcholine in the hippocampus and cortex of the brain and thus facilitating normal regeneration of synapses and functioning. Therefore, targeting both AChE and BuChE enzymes is one approach for treatment of AD [4,5,6,7]. Two binding sites are present in AChE: the peripheral site, which is responsible for beta amyloid interaction, and the catalytic site, which causes hydrolysis of acetylcholine. Interactions of beta amyloid protein (Aβ) with AChE access the formation of beta amyloid protein-acetylcholinesterase (Aβ-AChE) complex and thus result in neurotoxicity. BuChE is found in liver, intestine, heart, kidney, serum and lungs, while AChE is present in cholinergic neurons, brain and muscle [8,9]. The cholinesterase enzymes perform key roles in the breakdown of compounds, having ester moieties in their core structures. Generally, AChE is dominant in brain, while BuChE functions when acetylcholine gradually decreases its function in the brain of AD patients. Hence, synthesis of drugs that function as inhibitors of both AChE and BuChE enzymes should be effective treatments of AD [10]. Several drugs have been approved by the Food and Drug Administration (FDA) for treatment of AD. These include galantamine and donepezil which are selective for AChE, while rivastigmine and tacrine inhibit both BuChE and AChE (Figure 1) [11].

Triazole analogues are reported to have therapeutic and biological activities, such as anticonvulsant [12], anti-inflammatory [13], antifungal [14], insecticidal [15] and plant growth regulation [16]. There are some important drugs, such as letrozole (anticancer), tazobactam (antibacterial), isavuconazole (antifungal), sitagliptin (antidiabetic), ribavirin (antiviral), and rufinamide (seizure disorder), that contain triazole moieties in their core structures (Figure 2) [17,18,19,20,21,22,23].

Recently, several classes of heterocyclic compounds have been reported to be potent inhibitors of acetylcholinesterase (AChE) and butyrylcholinesterase (BuChE) [24,25,26,27,28]. Based on the biological importance of thiosemicarbazone [29,30] and triazole [31,32] compounds (Figure 3), it was decided to synthesize hybrid analogues based on triazole bearing thiosemicarbazone moiety as effective inhibitors of cholinesterase enzymes, such as AChE and BuChE, that could be effective treatments for AD.

## 2. Results and Discussion

### 2.1. Chemistry

Thiosemicarbazide (**1**) was treated with 4-nitrobenzoyl chloride in DMF in the presence of triethylamine and refluxed for 3 h to yield 2-(4-nitrobenzoyl)hydrazine-1-carbothioamide as the first intermediate (**2**), which further underwent cyclization during stirring overnight in 2% aqueous solution of sodium hydroxide followed by neutralization with dil. HCl to yield 1,2,4-triazole-3-thiole (**3**) as the second intermediate product. Intermediate (**3**) was then reacted with different substituted phenacyl bromide in ethanol in the presence of triethylamine and refluxed for 3 h to obtain the third intermediate (**4**). Intermediate (**4**) was then mixed with hydrazine hydrate in methanol in the presence of a few drops of glacial acetic acid to form intermediate (**5**). Finally, intermediate (**5**) was treated with different substituted isothiocyanate in tetrahydrofuran in the presence of triethylamine, with the resulting mixture stirred under reflux until conversion had been completed, as monitored by TLC during refluxing for 6–8 h. After being cooled to room temperature, the product was reacted with 5% Na_2_S_2_O_3_ and extracted with CH_2_Cl_2_/MeOH (10:1, 10 mL × 4). The combined organic layer was dried over anhydrous sodium sulfite and concentrated. The resulting residue was purified through silica gel column chromatography using a mixture of petroleum ether and EtOAc as eluent to yield the desired triazole-based thiosemicarbazone derivatives (**6a–u**) as the final product (Figure 1, Table 1). Primary confirmation of the product was done using thin layer chromatography (TLC) and further confirmed by with nuclear magnetic resonance (NMR).

### 2.2. In Vitro Inhibition of Acetylcholinesterase and Butyrylcholinesterase Activities

All the newly synthesized derivatives of triazole-based thiosemicarbazone (**6a–u**) were screened in vitro for inhibition of acetylcholinesterase (AChE) and butyrylcholinesterase (BuChE) activities. All the newly afforded derivatives, except **6c** and **6d**, which are found to be inactive, displayed good to moderate inhibition, with IC_50_ ranging from 0.10 ± 0.050 to 12.20 ± 0.30 µM against AChE and 0.20 ± 0.050 µM to 14.10 ± 0.40 µM against BuChE compared to the standard drug donepezil, which exhibited IC_50_ of 2.16 ± 0.12 and 4.5 ± 0.11 µM against AChE and BuChE, respectively (Table 1). A structure-activity relationship (SAR) based on substituent/s and electron donating/withdrawing effects on phenyl rings B and C was developed. The compounds were divided into five major parts: triazole moiety, ring A, thiosemicarbazone moiety, ring B, and ring C. Each part of the synthesized compounds was found to be actively participating in inhibition of both acetylcholinesterase (AChE) and butyrylcholinesterase (BuChE). Furthermore, it was determined that by keeping the triazole, ring A, and thiosemicarbazone moieties constant, the variation in inhibitory potentials was determined by attachment of substituents of diverse nature at various positions in different number/s around both rings B and C (Figure 4, Table 1).

#### Structure–Activity Relationship (SAR) for Inhibition of Acetylcholinesterase (AChE) and Butyrylcholinesterase (BuChE)

Analogue **6i** with IC_50_ = 0.10 ± 0.050 (for AChE) and IC_50_ = 0.20 ± 0.050 µM (for BuChE) having di-substitutions (2-nitro, 3-hydroxy groups) at ring **B** and tri-substitutions (2-nitro, 4,5-dichloro groups) at ring **C** emerged as the most active inhibitor of targeted AChE and BuChE enzymes, whereas analogue **6b** (IC_50_ = 0.20 ± 0.10 µM for AChE) (IC_50_ = 0.30 ± 0.10 µM for BuChE) having di-Cl at 4,5- and—NO_2_ groups at 2-position of phenyl ring **B** and hydroxy group at *ortho*-position of phenyl ring **C** was recognized as the second-most active among the current synthesized series (Table 1). The greater number of attached electron-withdrawing groups, such as di-Cl and -NO_2_ groups around ring C, as well as the presence of substituents (-OH) capable of forming hydrogen bonds with the active residue of amino acids of these analogues were responsible for enhanced inhibitory potentials for both targeted AChE and BuChE. The majority of the electronic density is removed from Ph-ring B and C by these di-Cl and -NO_2_ groups, making it electron-deficient and further regaining stability through interactions with the active sites of targeted AChE and BuChE enzymes. The derivative **6k**, which has a *para*-bromo substitution on ring B and a *para*-tolyl group at 4-position of aryl ring C, was shown to have the least inhibitory activity against AChE and BuChE enzymes with IC_50_ values of 12.20 ± 0.30 and 14.10 ± 0.40µM. This reduced the potency of derivative **6k**, caused by the greater size of the attached substituent(s), which increased the crowdedness around both rings B and C and thus reduced the chance of interactions with catalytic residues of targeted enzymes (Table 1).

Derivatives **6a**, **6b** and **6l** containing di-Cl groups at meta- and para-position and nitro group at ortho-position of ring C and a variety of other groups, including Br, OH, NO_2_ and CH_3_, at various position of ring B, improved inhibition of activities of both AChE and BuChE enzymes. Among these three derivatives, derivative **6b** (IC_50_ = 0.20 ± 0.10 and 0.30 ± 0.10 µM) with hydroxy group at ortho-position on ring B along with tri-substitutions (2-nitro and 3,4-di-Cl groups) at aryl ring C displayed superior inhibition of AChE and BuChE, compared to derivative **6a** (IC_50_ = 5.10 ± 0.20 and 6.40 ± 0.20 µM), which had a bromo group at ortho-position of ring B and derivative **6l** (IC_50_ = 4.60 ± 0.010 and 5.90 ± 0.10 µM), bearing NO_2_ at ortho and CH_3_ groups at para on ring B along with di-Cl groups at meta- and para-position and nitro group at ortho-position of ring C (Table 1). These three derivatives contain tri-substitutions (2-nitro, 3,4-dichloro groups) at ring C, but have different substituents (Br, OH, CH3 and NO_2_ groups) around ring B. This diverse nature of substituents around ring B have different tendencies to interact with active site of targeted enzymes and hence cause variation in inhibitory potentials of these three derivatives (Table 1). Moreover, derivative **6a** which contains a bromo at ortho on ring B and di-Cl groups at meta- and para-position and nitro group at ortho-position of ring C, exhibits better inhibition of AChE and BuChE activities than derivative **6h**, which has a bromo moiety at para on ring B and ortho-nitro and para-methyl substitutions on ring C, might be due to di-Cl groups on ring, as well as different position of bromo moiety around ring B (Table 1).

Derivative **6g** (IC_50_ = 2.10 ± 0.10 and 4.30 ± 0.10 µM), containing two methoxy groups at ortho- and meta-position of ring B and a nitro group at the para-position of the phenyl ring C, exhibited better inhibition of AChE and BuChE activities, compared to derivative **6f** (IC_50_ = 2.40 ± 0.10 and 4.70 ± 0.10 µM), which had two methoxy groups at the ortho- and meta-positions of ring B and ortho-nitro and para-methyl groups on phenyl ring C (Table 1). This enhanced inhibition of AChE and BuChE activities of derivative **6g** might be due to the stronger electron withdrawing nature of nitro group, making the ring C partially positive, which further established a pi-cation interaction with the active enzyme site. Alternatively, derivative **6f** had both electron-donating (CH_3_) and electron-withdrawing (NO_2_) groups on ring C, which did not create charge on ring C and hence resulted in lesser activities of AChE and BuChE (Table 1).

Comparing derivative **6h** (IC_50_ = 9.10 ± 0.20 and 11.20 ± 0.30 µM), which has a bromo group at the para position of the phenyl ring B and a nitro group at ortho-position and a methyl group at the para-position of the phenyl ring C, with derivatives **6m** (IC_50_ = 2.70 ± 0.10 and 3.80 ± 0.10 µM) having di-Cl groups at meta- and para-positions of ring B and nitro group at ortho-position and methyl group at para-position on phenyl ring C and **6r** (IC_50_ = 1.30 ± 0.050 and 2.20 ± 0.10 µM) having nitro group at ortho on ring B and nitro group at ortho-position and methyl group at para-position on phenyl ring C and **6t** (IC_50_ = 1.90 ± 0.10 and 2.50 ± 0.10 µM) with a methoxy group at meta-position of ring B and the nitro group at ortho-position and the methyl group at para-position on phenyl ring C (Table 1). The small difference in the inhibitory activities (AChE and BuChE) of all derivatives might be due to the different nature and position of the substituent/s on phenyl ring B (Table 1).

Derivative **6n** (IC_50_ = 0.70 ± 0. 05 and 1.70 ± 0.050 µM), which has di-chloro groups at meta- and para-positions on ring B and a nitro group at para-position on ring C, with derivative **6s** (IC_50_ = 1.40 ± 0.050 and 2.30 ± 0.10µM), which has a nitro group at ortho position on ring B and nitro moiety at para position on ring C, and derivative **6u** (IC_50_ = 2.90 ± 0.10 and 3.70 ± 0.10 µM), with a methoxy group at the meta-position on ring B and nitro moiety at the para position on ring C were compared (Table 1). Difference in the inhibitory activities (AChE and BuChE) of these derivatives might be due to the different nature and position of substituent/s on ring B (Table 1). Number, nature, position and electron-donating/withdrawing nature of substituents considerably influenced inhibition of activities.

### 2.3. Docking Study

Molecular docking was analyzed in order to gain an understanding of the binding mechanism of synthesized compounds against both the targeted enzymes. The optimized compounds were docked based on the co-crystal of each crystallographic structure. Each compound was assigned a total of 30 conformations prior to the docking process. For further investigation, the top-ranked conformations of potent compounds were chosen. The docking results revealed that all the compounds were well oriented in the active site of both enzymes. In general, we found that all of the compounds in the series—with different substituted groups at all three ends of the compound (according to the scheme), where one end has a nitro group, the second end has a halogen group, and the most important end (third) has a different substituted group—had inhibitory potential against the target. These typically belong to electron-withdrawing or electron-donating groups.

The protein–ligand interaction profile of the most active analogue, **6i**, had several key interactions with catalytic residues of acetylcholinesterase enzyme, including Tyr332 (pi-sigma and pi-pi stacked), Trp82 (pi-anion), Glu197 (HB), Phe329 (pi-sulfur), Ser198 (HB), His438 (HB), Asn289 (HB), Asn68 (HB) and Asp70 (pi-cation) (Figure 5), while against butyrylcholinesterase enzymes, it interacts through numerous interactions such as Tyr116 (pi-alkyl), Tyr130 (pi-alkyl), His440 (CHB), Phe330 (pi-anion), Asp72 (pi-cation), Asn85 (HB), Tyr121 (pi-sulfur), Trp279 (pi-pi shaped and amide-pi stacked), Tyr70 (HB), Tyr334 (pi-pi stacked), Trp84 (pi-pi T shaped stacked), Gly118 (HB) and Gly117 (pi-pi stacked) (Figure 6).

As for the second-most active analogue, **6b**, the protein–ligand interaction (PLI) profile showed that this scaffold enhanced inhibitory potential against AChE by interacting with the active part of the AChE enzyme through several significant interactions, such as Gly118 (CHB), Gly117 (pi-pi stacked), Ile444 (pi-alkyl), Tyr130 (pi-alkyl), Trp84 (pi-sigma and pi-pi stacked), Phe330 (pi-donor HB), His440 (pi-alkyl), Tyr121 (HB), Tyr334 (pi-sulfur and amide-pi stacked), Tyr70 (HB and pi-pi T shaped), Trp279 (pi-pi T shaped), Ser122 (HB) and Asp72 (pi-anion) (Figure 7), while against butyrylcholinesterase this analogue **6b** adopted numerous key interactions including Pro285 (HB), His438 (pi-alkyl), Phe329 (pi-sulfur and pi-alkyl), Ala328 (alkyl), Gly117 (HB), Gly116 (HB), Thr120 (pi-sigma), Trp82 (pi-pi stacked), Gly115 (HB) and Tyr128 (HB) (Figure 8).

The protein–ligand interaction (PLI) profile of the third-most active analogue, **6q**, showed that it established numerous interesting key interactions with active sites of the acetylcholinesterase enzyme, including Tyr128 (HB), Trp82 (pi-pi stacked), Trp430 (pi-alkyl), Tyr332 (pi-alkyl), Ala328 (pi-alkyl), pro285 (HB), Phe329 (pi-pi T shaped and pi-alkyl), Leu286 (pi-alkyl), Trp231 (pi-pi T shaped), Gly117 (HB), Ser198 (HB), Gly116 (HB), His438 (HB and pi-alkyl) and Asp70 (pi-anion) (Figure 9), while against the butyrylcholinesterase enzyme it established several important interactions, such as Tyr130 (HB), Tyr70 (pi-sulfur), Tyr338 (pi-pi T shaped), Arg289 (alkyl), Phe331 (pi-pi stacked and pi-sulfur), Trp279 (pi-pi stacked), Trp84 (pi-stacked), Phe330 (pi-pi stacked) and Gly117 (HB) as shown in (Figure 10).

Calculated binding energies, number of hydrogen bonds, and the number of closest residues surround the selected docked analogues into the active site of both AChE and BuChE enzymes are shown in Table 2.

## 3. Experimental

All chemicals and solvents were purchased from Sigma Aldrich (St. Louis, MO, USA) with a purity of 97 up to 99%.

### 3.1. General Procedure of 1,2,4-Triazole Bearing Thiosemicarbazone Derivatives (***6a–u***)

Thiosemicarbazide (**1**, 0.5 mmol) was treated with 4-nitrobenzoyl chloride (0.5 mmol) in DMF (10 mL) in the presence of triethylamine (0.5 mL) and refluxed for 3 h to yield 2-(4-nitrobenzoyl)hydrazine-1-carbothioamide as first intermediate (**2**), which further undergoes cyclization on stirring overnight in 2% aqueous solution of sodium hydroxide (10 mL followed by neutralization with dilute HCl (5 mL) to yield 1,2,4-triazole-3-thiole (**3**) as the second intermediate product. Intermediate (**3**) was then reacted with equivalent different substituted phenacyl bromide in ethanol (10 mL) in the presence of triethylamine (0.5 mL) and refluxed for 3 h to obtain third intermediate (**4**). Intermediate (**4**) was then mixed with hydrazine hydrate (5 mL) in methanol (10 mL) in the presence of a few drops of glacial acetic acid to give fourth intermediate (**5**). Finally intermediate (**5**) was treated with equivalent different substituted isothiocyanate in tetrahydrofuran (10 mL) in the presence of triethylamine (0.5 mL) and the resulting mixture was stirred under reflux until the conversion was completed (monitored by TLC, reflux 6–8 h). After being cooled to room temperature, it was reacted with 5% Na_2_S_2_O_3_ (20 mL) and extracted with CH_2_Cl_2_/MeOH (10:1, 10 mL × 4). The combined organic layer was dried over anhydrous sodium sulfate and concentrated. The given residue was purified through silica gel column chromatography using a mixture of petroleum ether and EtOAc as eluent to yield the desired triazole-based thiosemicarbazone derivatives (**6a–u**).

### 3.2. Spectral Analysis

All the proton NMR spectra are shown in Appendix A.

#### 3.2.1. (.E)-2-(1-(2-bromophenyl)-2-((5-(4-nitrophenyl)-4H-1,2,4-triazol-3-yl)thio)ethylidene)-N-(4,5-dichloro-2-nitrophenyl)hydrazine-1-carbothioamide (**6a**)

Yield: 63%; ^1^H NMR (600 MHz, DMSO-*d_6_*): *δ* 13.25 (s, 1H, NH), 11.81 (s, 1H, NH), 10.11 (s, 1H, NH), 8.28 (s, 1H, Ar-H), 8.10 (s, 1H, Ar-H), 7.88 (d, *J* = 8.88 Hz, 2H, Ar-H), 7.79 (d, *J* = 8.46 Hz, 2H, Ar-H), 7.43 (t, *J* = 7.92 Hz, 2H, Ar-H), 7.27 (t, *J* = 7.68 Hz, 1H, Ar-H), 7.04 (d, *J* = 7.56 Hz, 1H, Ar-H), 2.34 (s, 2H, S-CH_2_). ^13^C NMR (150 MHz, DMSO-*d_6_*): *δ* 184.0, 158.5, 157.3, 155.4, 147.5, 146.2, 138.4, 137.1, 135.2, 134.4, 132.5, 131.6, 130.0, 129.0, 127.5, 126.8, 126.8, 126.3, 125.5, 124.1, 124.1, 122.0, 31.0. HRMS (ESI) *m*/*z*: [M+H]^+^ calcd for C_23_H_16_BrCl_2_N_8_O_4_S_2_, 680.8117; Found, 680.8100.

#### 3.2.2. (.E)-N-(4,5-dichloro-2-nitrophenyl)-2-(1-(2-hydroxyphenyl)-2-((5-(4-nitrophenyl)-4H-1,2,4-triazol-3-yl)thio)ethylidene)hydrazine-1-carbothioamide (**6b**)

Yield: 72%; ^1^HNMR (600 MHz, DMSO-*d_6_*): *δ* 13.39 (s, 1H, NH), 13.27 (s, 1H, NH), 12.01 (s, 1H, NH), 10.15 (s, 1H, OH), 8.91 (s, 1H, Ar-H), 8.30 (s, 1H, Ar-H), 8.16-7.87 (m, 4H, Ar-H), 7.83-7.72 (m, 2H, Ar-H), 7.45 (t, *J* = 7.74 Hz, 1H, Ar-H), 7.25 (t, *J* = 7.74 Hz, 1H, Ar-H), 2.28 (s, 2H, S-CH_2_). ^13^C NMR (150 MHz, DMSO-*d_6_*): *δ* 183.9, 162.5, 158.4, 157.2, 155.3, 147.4, 146.4, 138.3, 137.2, 132.4, 132.1, 131.5, 129.2, 127.1, 127.1, 126.4, 125.5, 124.1, 124.1, 121.0, 118.3, 117.0, 32.0. HRMS (ESI) *m*/*z*: [M+H]^+^ calcd for C_23_H_17_Cl_2_N_8_O_5_S_2_,619.0150; Found, 619.0134.

#### 3.2.3. (.E)-2-(1-([1,1′-biphenyl]-4-yl)-2-((5-(4-nitrophenyl)-4H-1,2,4-triazol-3-yl)thio)ethylidene)-N-phenylhydrazine-1-carbothioamide (**6c**)

Yield: 75%; ^1^H NMR (600 MHz, DMSO-*d_6_*): *δ* 13.36 (s, 1H, -NH), 11.25 (s, 1H, -NH), 11.08 (s, 1H, -NH), 8.23 (d, *J* = 9.1 Hz, 2H, Ar-H), 7.99 (d, *J* = 9.1 Hz, 2H, Ar-H), 7.96 (d, *J* = 8.4 Hz, 2H, Ar-H), 7.86 (dd, *J* = 7.3 Hz, 2H, Ar-H), 7.72 (dd, *J* = 7.8 Hz, 1.9 Hz, 2H, Ar-H), 7.63-7.56 (m, 5H, Ar-H), 7.40-7.35 (m, 2H, Ar-H), 7.06-6.98 (m, 1H, Ar-H), 3.73 (s, 2H, S-CH_2_). ^13^C NMR (150 MHz, DMSO-*d_6_*): *δ* 183.8, 158.4, 157.2, 155.2, 147.5, 142.7, 140.4, 138.2, 138.1, 132.5, 129.3, 129.3, 128.8, 128.8, 128.6, 128.6, 128.0, 127.6, 127.6, 127.5, 127.5, 127.2, 126.6, 126.6, 126.1, 126.1, 124.0, 124.0, 31.4. HRMS (ESI) *m*/*z*: [M+H]^+^ calcd for C_29_H_24_N_7_O_2_S_2_,566.1429; Found, 566.1408.

#### 3.2.4. (.E)-2-(1-([1,1′-biphenyl]-4-yl)-2-((5-(4-nitrophenyl)-4H-1,2,4-triazol-3-yl)thio)ethylidene)-N-(4′-methyl-[1,1′-biphenyl]-4-yl)hydrazine-1-carbothioamide (**6d**)

Yield: 73%; ^1^ H NMR (600 MHz, DMSO-*d_6_*): *δ* 13.38 (s, 1H, -NH), 11.24 (s, 1H, -NH), 11.07 (s, 1H, -NH), 8.22 (d, *J* = 6.8 Hz, 2H, Ar-H), 7.98 (d, *J* = 8.6 Hz, 2H, Ar-H), 7.95 (d, *J* = 7.2 Hz, 2H, Ar-H), 7.85 (dd, *J* = 7.8 Hz, 2H, Ar-H), 7.78-7.69 (m, 5H, Ar-H), 7.67 (d, *J* = 7.6 Hz, 2H, Ar-H), 7.63 (d, *J* = 7.3 Hz, 2H, Ar-H), 7.30 (d, *J* = 6.6 Hz, 2H, Ar-H), 7.12 (d, *J* = 6.7 Hz, 2H, Ar-H), 3.71 (s, 2H, S-CH_2_), 2.31 (s, 3H, -CH_3_). ^13^C NMR (150 MHz, DMSO-*d_6_*): *δ* 183.7, 158.3, 57.1, 155.1, 147.4, 142.6, 140.3, 139.3, 138.1, 137.3, 136.9, 132.4, 131.3, 130.1, 129.2, 129.2, 129.0, 129.0, 128.7, 128.7, 127.7, 127.7, 127.5, 127.5, 127.4, 127.4, 127.3, 127.3, 127.1, 126.5, 126.5, 124.1, 124.1, 123.9, 123.9, 20.8.; HRMS (ESI) *m*/*z*: [M+H]^+^ calcd for C_36_H_30_N_7_O_2_S_2_,656.1898; Found, 656.1877.

#### 3.2.5. (.E)-2-(1-([1,1′-biphenyl]-4-yl)-2-((5-(4-nitrophenyl)-4H-1,2,4-triazol-3-yl)thio)ethylidene)-N-(3-nitrophenyl)hydrazine-1-carbothioamide (**6e**)

Yield: 83%; ^1^H NMR (600 MHz, DMSO-*d_6_*): *δ* 13.35 (s, 1H, -NH), 11.31 (s, 1H, -NH), 11.14 (s, 1H, -NH), 8.58 (dd, *J* = 2.4 Hz, 2.0 Hz, 1H, Ar-H), 8.29 (d, *J* = 9.0 Hz, 2H, Ar-H), 8.05 (d, *J* = 9.2 Hz, 2H, Ar-H), 8.02 (d, *J* = 8.5 Hz, 2H, Ar-H), 7.94 (dd, *J* = 7.5 Hz, 2H, Ar-H), 7.92–7.87 (m, 1H, Ar-H), 7.85–7.79 (m, 1H, Ar-H), 7.75–7.63 (m, 5H, Ar-H), 7.60 (t, *J* = 8.6 Hz, 1H, Ar-H), 3.79 (s, 2H, S-CH_2_). ^13^C NMR (150 MHz, DMSO-*d_6_*): *δ* 184.0, 158.6, 157.4, 155.4, 148.0, 147.7, 142.9, 140.6, 137.8, 136.4, 132.7, 132.4, 129.7, 129.5, 129.5, 129.0, 129.0, 127.8, 127.8, 127.7, 127.7, 127.4, 126.8, 126.8, 124.2, 124.2, 119.7, 119.2, 31.6. HRMS (ESI) *m*/*z*: [M+H]^+^ calcd for C_29_H_23_N_8_O_4_S_2_,611.1176; Found, 611.1159.

#### 3.2.6. (.E)-2-(1-(2,5-dimethoxyphenyl)-2-((5-(4-nitrophenyl)-4H-1,2,4-triazol-3-yl)thio)ethylidene)-N-(4-methyl-2-nitrophenyl)hydrazine-1-carbothioamide (**6f**)

Yield: 80%; ^1^H NMR (600 MHz, DMSO-*d_6_*): *δ* 13.33 (s, 1H, -NH), 11.15 (s, 1H, -NH), 10.81 (s, 1H, -NH), 8.85 (d, *J* = 6.9 Hz, 1H, Ar-H), 8.33 (d, *J* = 2.1 Hz, 1H, Ar-H), 8.28 (d, *J* = 9.3 Hz, 2H, Ar-H), 8.05 (d, *J* = 9.0 Hz, 2H, Ar-H), 7.42 (d, *J* = 2.0 Hz, 1H, Ar-H), 7.06 (d, *J* = 7.1 Hz, 1H, Ar-H), 7.98 (dd, *J* = 7.1 Hz, 1.5 Hz, 1H, Ar-H), 7.01 (dd, *J* = 7.9 Hz, 1.7 Hz, 1H, Ar-H), 3.74 (s, 2H, S-CH_2_), 2.41 (s, 3H, CH_3_), 3.92 (s, 3H, -OCH_3_), 3.79 (s, 3H, -OCH_3_). ^13^C NMR (150 MHz, DMSO-*d_6_*): *δ* 184.1, 158.7, 157.5, 155.5, 152.9, 152.6, 147.8, 147.1, 138.5, 135.3, 135.2, 129.3, 126.9, 126.9, 125.2, 124.6, 124.2, 124.2, 118.1, 117.5, 115.3, 114.1, 32.0, 58.6, 58.3, 20.2. HRMS (ESI) *m*/*z*: [M+H]^+^ calcd for C_26_H_25_N_8_O_6_S_2_,609.1334; Found, 609.1320.

#### 3.2.7. (.E)-2-(1-(2,5-dimethoxyphenyl)-2-((5-(4-nitrophenyl)-4H-1,2,4-triazol-3-yl)thio)ethylidene)-N-(4-nitrophenyl)hydrazine-1-carbothioamide (**6g**)

Yield: 80%; ^1^H NMR (600 MHz, DMSO-*d_6_*): *δ* 13.36 (s, 1H, -NH), 11.16 (s, 1H, -NH), 10.80 (s, 1H, -NH), 8.27 (d, *J* = 9.1 Hz, 2H, Ar-H), 8.18 (d, *J* = 6.9 Hz, 2H, Ar-H), 8.04 (d, *J* = 9.3 Hz, 2H, Ar-H), 7.74 (d, *J* = 6.8 Hz, 2H, Ar-H), 7.41 (d, *J* = 2.2 Hz, 1H, Ar-H),7.07 (d, *J* = 7.2 Hz, 1H, Ar-H), 7.02 (dd, *J* = 7.8 Hz, 1.8 Hz, 1H, Ar-H), 3.73 (s, 2H, S-CH_2_), 3.91 (s, 3H, -OCH_3_), 3.78 (s, 3H, -OCH_3_). ^13^C NMR (150 MHz, DMSO-*d_6_*): *δ* 184.4, 159.0, 157.8, 155.8, 153.2, 152.9, 148.1, 144.8, 144.1, 138.8, 127.2, 127.2, 125.0, 125.0, 124.6, 124.6, 124.4, 124.4, 118.4, 117.8, 115.6, 114.5, 56.0, 55.6, 32.3. HRMS (ESI) *m*/*z*: [M+H]^+^ calcd for C_25_H_23_N_8_O_6_S_2_,595.1179; Found, 595.1157.

#### 3.2.8. (.E)-2-(1-(4-bromophenyl)-2-((5-(4-nitrophenyl)-4H-1,2,4-triazol-3-yl)thio)ethylidene)-N-(4-methyl-2-nitrophenyl)hydrazine-1-carbothioamide (**6h**)

Yield: 79%; ^1^H NMR (600 MHz, DMSO-*d_6_*): *δ* 13.32 (s, 1H, -NH), 11.06 (s, 1H, -NH), 10.72 (s, 1H, -NH), 8.78 (d, *J* = 7.0 Hz, 1H, Ar-H), 8.28 (d, *J* = 1.8 Hz, 1H, Ar-H), 8.25 (d, *J* = 9.5 Hz, 2H, Ar-H), 8.01 (d, *J* = 7.3 Hz, 2H, Ar-H), 7.73 (d, *J* = 7.6 Hz, 2H, Ar-H), 7.91 (dd, *J* = 8.3 Hz, 1.5 Hz, 1H, Ar-H), 7.67 (dd, *J* = 7.9 Hz, 2H, Ar-H), 3.73 (s, 2H, S-CH_2_), 2.40 (s, 3H, CH_3_). ^13^C NMR (150 MHz, DMSO-*d_6_*): *δ* 183.9, 158.5, 157.3, 155.3, 147.6, 146.9, 138.3, 135.1, 135.0, 132.7, 131.4, 131.4, 129.1, 128.3, 128.3, 126.7, 126.7, 125.1, 125.0, 124.3, 124.1, 124.1, 31.5, 20.1. HRMS (ESI) *m*/*z*: [M+H]^+^ calcd for C_24_H_20_BrN_8_O_4_S_2_,627.0227; Found, 627.0217.

#### 3.2.9. (.E)-N-(4,5-dichloro-2-nitrophenyl)-2-(1-(3-hydroxy-2-nitrophenyl)-2-((5-(4-nitrophenyl)-4H-1,2,4-triazol-3-yl)thio)ethylidene)hydrazine-1-carbothioamide (**6i**)

Yield: 61%; ^1^H NMR (600 MHz, DMSO-*d_6_*): *δ* 13.39 (s, 1H, NH), 13.30 (s, 1H, NH), 12.21 (s, 1H, NH), 10.51 (s, 1H, -OH), 8.91 (s, 1H, Ar-H), 8.35 (s, 1H, Ar-H), 8.30 (d, *J* = 9.62 Hz, 1H, Ar-H), 8.27–8.03 (m, 1H, Ar-H), 7.91–7.85 (m, 1H, Ar-H), 7.83 (d, *J* = 9.0 Hz, 2H, Ar-H), 7.76 (d, *J* = 8.52 Hz, 2H, Ar-H), 2.34 (s, 2H, S-CH_2_). ^13^C NMR (150 MHz, DMSO-*d_6_*): *δ* 184.4, 158.3, 157.4, 155.2, 153.0, 148.6, 147.5, 139.2, 138.4, 137.1, 136.5, 136.0, 131.5, 129.0, 126.6, 126.6,126.3, 125.5, 124.0, 124.0, 121.4, 117.9, 30.4. HRMS (ESI) *m*/*z*: [M+H]^+^ calcd for C_23_H_15_Cl_2_N_9_O_7_S_2_, 662.9611; Found, 664.9600.

#### 3.2.10. (.E)-2-(1-([1,1′-biphenyl]-4-yl)-2-((5-(4-nitrophenyl)-4H-1,2,4-triazol-3-yl)thio)ethylidene)-N-(2,5-dichloro-4-(dimethylamino)phenyl)hydrazine-1-carbothioamide (**6j**)

Yield: 64%; ^1^H NMR (600 MHz, DMSO-*d_6_*): *δ* 13.64 (s, 1H, NH), 11.54 (s, 1H, NH), 10.12 (s, 1H, NH), 8.23 (s, 1H, Ar-H), 8.18 (s, 1H, Ar-H), 8.04 (d, *J* = 7.26 Hz, 4H, Ar-H), 7.99–7.92 (m, 4H, Ar-H), 7.64–7.59 (m, 5H, Ar-H), 4.03 (s, 6H, -CH_3_), 2.51 (s, 2H, S-CH_2_). ^13^C NMR (150 MHz, DMSO-*d_6_*): *δ* 184.0, 158.5, 157.3, 155.2, 148.3, 147.5, 143.0, 140.6, 138.4, 134.2, 132.5, 129.6, 129.6, 129.0, 128.9, 128.9, 127.8, 127.8, 127.4, 127.4, 127.1, 127.1, 126.9, 126.9, 126.4, 124.2, 124.2, 120.6, 40.5, 40.5, 31.4. HRMS (ESI) *m*/*z*: [M+H]^+^ calcd for C_31_H_27_Cl_2_N_8_O_2_S_2_, 678.0719; Found, 678.0701.

#### 3.2.11. (.E)-2-(1-(4-bromophenyl)-2-((5-(4-nitrophenyl)-4H-1,2,4-triazol-3-yl)thio)ethylidene)-N-(4′-methyl-[1,1′-biphenyl]-4-yl)hydrazine-1-carbothioamide (**6k**)

Yield: 73%; ^1^H NMR (600 MHz, DMSO-*d_6_*): *δ* 13.31 (s, 1H, -NH), 11.25 (s, 1H, -NH), 11.08 (s, 1H, -NH), 8.21 (d, *J* = 6.9 Hz, 2H, Ar-H), 7.97 (d, *J* = 8.9 Hz, 2H, Ar-H), 7.73 (d, *J* = 7.3 Hz, 2H, Ar-H), 7.59 (dd, *J* = 8.8 Hz, 2H, Ar-H), 7.68 (d, *J* = 7.7 Hz, 2H, Ar-H), 7.64 (d, *J* = 7.4 Hz, 2H, Ar-H), 7.31 (d, *J* = 6.7 Hz, 2H, Ar-H), 7.13 (d, *J* = 6.8 Hz, 2H, Ar-H), 3.72 (s, 2H, S-CH_2_), 2.32 (s, 3H, -CH_3_). ^13^C NMR (150 MHz, DMSO-*d_6_*): *δ* 184.0, 158.6, 157.4, 155.4, 147.7, 139.6, 138.4, 137.6, 137.2, 132.8, 131.5, 131.5, 130.4, 129.3, 129.3, 128.4, 128.4, 128.0, 128.0, 127.6, 127.6, 126.8, 126.8, 125.2, 124.4, 124.4, 124.2, 124.2, 31.6, 21.1. HRMS (ESI) *m*/*z*: [M+H]^+^ calcd for C_30_H_25_BrN_7_O_2_S_2_,658.0689; Found, 658.0669.

#### 3.2.12. (.E)-N-(4,5-dichloro-2-nitrophenyl)-2-(1-(4-methyl-2-nitrophenyl)-2-((5-(4-nitrophenyl)-4H-1,2,4-triazol-3-yl)thio)ethylidene)hydrazine-1-carbothioamide (**6l**)

Yield: 67%; ^1^H NMR (600 MHz, DMSO-*d_6_*): *δ* 13.62 (s, 1H, NH), 13.38 (s, 1H, NH), 10.12 (s, 1H, NH), 8.90 (s, 1H, Ar-H), 8.23 (s, 1H, Ar-H), 8.16 (s, 1H, Ar-H), 8.02 (d, *J* = 5.58 Hz, 1H, Ar-H), 7.87 (d, *J* = 8.46 Hz, 1H, Ar-H), 7.75 (d, *J* = 8.4 Hz, 2H, Ar-H), 7.61 (d, *J* = 8.34 Hz, 2H, Ar-H), 2.50 (s, 2H, S-CH_2_), 1.91 (s, 3H, -CH_3_). ^13^C NMR (150 MHz, DMSO-*d_6_*): *δ* 184.0, 158.6, 157.4, 155.3, 147.5, 147.2, 146.5, 141.4, 138.4, 137.0, 135.1, 132.2, 131.6, 129.9, 129.0, 126.8, 126.8, 126.3, 125.4, 125.2, 124.2, 124.2, 30.5, 20.0. HRMS (ESI) *m*/*z*: [M+H]^+^ calcd for C_24_H_18_Cl_2_N_9_O_6_S_2_,662.0130; Found, 662.0111.

#### 3.2.13. (.E)-2-(1-(3,4-dichlorophenyl)-2-((5-(4-nitrophenyl)-4H-1,2,4-triazol-3-yl)thio)ethylidene)-N-(4-methyl-2-nitrophenyl)hydrazine-1-carbothioamide (**6m**)

Yield: 61%; ^1^H NMR (600 MHz, DMSO-*d_6_*): *δ* 13.43 (s, 1H, -NH), 11.06 (s, 1H, -NH), 10.71 (s, 1H, -NH), 8.79 (d, *J* = 7.1 Hz, 1H, Ar-H), 8.29 (d, *J* = 1.7 Hz, 1H, Ar-H), 8.24 (d, *J* = 9.4 Hz, 2H, Ar-H), 8.02 (d, *J* = 7.5 Hz, 2H, Ar-H), 7.90 (dd, *J* = 8.4 Hz, 1.2Hz, 1H, Ar-H), 7.83 (d, *J* = 1.5 Hz, 1H, Ar-H), 7.80 (dd, *J* = 8.5 Hz, 1.9 Hz, 1H, Ar-H), 7.66 (d, *J* = 8.9 Hz, 1H, Ar-H), 3.72 (s, 2H, S-CH_2_), 2.40 (s, 3H, CH_3_). ^13^C NMR (150 MHz, DMSO-*d_6_*): *δ* 184.5, 159.1, 157.9, 155.9, 148.2, 147.5, 138.9, 136.0, 135.7, 135.6, 134.1, 133.8, 130.9, 130.6, 129.7, 127.3, 127.3, 126.6, 125.6, 124.7, 124.7, 124.5, 32.1, 20.6; HRMS (ESI) *m*/*z*: [M+H]^+^ calcd for C_24_H_19_Cl_2_N_8_O_4_S_2_,617.0341; Found, 617.0327.

#### 3.2.14. (.E)-2-(1-(3,4-dichlorophenyl)-2-((5-(4-nitrophenyl)-4H-1,2,4-triazol-3-yl)thio)ethylidene)-N-(4-nitrophenyl)hydrazine-1-carbothioamide (**6n**)

Yield: 59%; ^1^H NMR (600 MHz, DMSO-*d_6_*): *δ* 13.30 (s, 1H, -NH), 11.21 (s, 1H, -NH), 11.01 (s, 1H, -NH), 8.28 (d, *J* = 8.8 Hz, 2H, Ar-H), 8.23 (d, *J* = 9.6 Hz, 2H, Ar-H), 8.03 (d, *J* = 7.8 Hz, 2H, Ar-H), 7.85 (d, *J* = 1.8 Hz, 1H, Ar-H), 7.81 (dd, *J* = 8.5 Hz, 2.7 Hz, 1H, Ar-H), 7.74 (d, *J* = 8.0 Hz, 2H, Ar-H), 7.68 (d, *J* = 9.0 Hz, 1H, Ar-H), 3.73 (s, 2H, S-CH_2_). ^13^C NMR (150 MHz, DMSO-*d_6_*): *δ* 184.5, 159.1, 157.9, 155.9, 148.2, 144.2, 144.9, 138.9, 136.0, 134.0, 133.8, 130.9, 130.6, 127.3, 127.3, 126.6, 124.7, 124.7, 124.5, 124.5, 125.1, 125.1, 32.1; HRMS (ESI) *m*/*z*: [M+H]^+^ calcd for C_23_H_17_Cl_2_N_8_O_4_S_2_,603.0183; Found, 603.0169.

#### 3.2.15. (.E)-2-(1-(3,4-dichlorophenyl)-2-((5-(4-nitrophenyl)-4H-1,2,4-triazol-3-yl)thio)ethylidene)-N-phenylhydrazine-1-carbothioamide (**6o**)

Yield: 62%; ^1^H NMR (600 MHz, DMSO-*d_6_*): *δ* 13.41 (s, 1H, -NH), 11.31 (s, 1H, -NH), 11.15 (s, 1H, -NH), 8.29 (d, *J* = 6.8 Hz, 2H, Ar-H), 7.97 (d, *J* = 6.4 Hz, 2H, Ar-H), 7.90 (dd, *J* = 8.6 Hz, 2.1 Hz, 1H, Ar-H), 7.88 (d, *J* = 1.9 Hz, 1H, Ar-H), 7.73 (d, *J* = 8.0 Hz, 1H, Ar-H), 7.70 (dd, *J* = 7.7 Hz, 1.5 Hz, 2H, Ar-H), 7.43–7.35 (m, 2H, Ar-H), 7.08–6.99 (m, 1H, Ar-H), 3.74 (s, 2H, S-CH_2_). ^13^C NMR (150 MHz, DMSO-*d_6_*): *δ* 184.4, 159.0, 157.8, 155.8, 148.1, 138.8, 138.7, 135.9, 133.9, 133.7, 130.8, 130.5, 129.2, 129.2, 128.6, 127.2, 127.2, 126.7, 126.7, 126.5, 124.6, 124.6, 32.0. HRMS (ESI) *m*/*z*: [M+H]^+^ calcd for C_23_H_18_Cl_2_N_7_O_2_S_2_,558.0337; Found, 558.0326.

#### 3.2.16. (.E)-2-(1-(3,4-dichlorophenyl)-2-((5-(4-nitrophenyl)-4H-1,2,4-triazol-3-yl)thio)ethylidene)-N-(4′-methyl-[1,1′-biphenyl]-4-yl)hydrazine-1-carbothioamide (**6p**)

Yield: 64%; ^1^H NMR (600 MHz, DMSO-*d_6_*): *δ* 13.42 (s, 1H, -NH), 11.23 (s, 1H, -NH), 11.06 (s, 1H, -NH), 8.21 (d, *J* = 7.9 Hz, 2H, Ar-H), 7.97 (d, *J* = 8.7 Hz, 2H, Ar-H), 7.67 (d, *J* = 7.7 Hz, 2H, Ar-H), 7.91 (dd, *J* = 8.5 Hz, 2.0 Hz, 1H, Ar-H), 7.89 (d, *J* = 1.8 Hz, 1H, Ar-H), 7.74 (d, *J* = 8.1 Hz, 1H, Ar-H), 7.64 (d, *J* = 7.4 Hz, 2H, Ar-H), 7.31 (d, *J* = 6.7 Hz, 2H, Ar-H), 7.13 (d, *J* = 6.8 Hz, 2H, Ar-H), 3.72 (s, 2H, S-CH_2_), 2.30 (s, 3H, -CH_3_). ^13^C NMR (150 MHz, DMSO-*d_6_*): *δ* 184.3, 158.6, 157.4, 155.4, 147.7, 139.6, 138.4, 137.6, 137.2, 135.4, 134.2, 133.3, 130.4, 130.2, 130.1, 129.3, 129.3, 128.0, 128.0, 127.6, 127.6, 126.8, 126.8, 126.1, 124.4, 124.4, 124.2, 124.2, 31.7, 21.1. HRMS (ESI) *m*/*z*: [M+H]^+^ calcd for C_30_H_24_Cl_2_N_7_O_2_S_2_,648.0806; Found, 648.0790.

#### 3.2.17. (.E)-2-(1-(3,4-dichlorophenyl)-2-((5-(4-nitrophenyl)-4H-1,2,4-triazol-3-yl)thio)ethylidene)-N-(3-nitrophenyl)hydrazine-1-carbothioamide (**6q**)

Yield: 67%; ^1^H NMR (600 MHz, DMSO-*d_6_*): *δ* 13.36 (s, 1H, -NH), 11.29 (s, 1H, -NH), 11.13 (s, 1H, -NH), 8.58 (dd, *J* = 2.3 Hz, 2.0 Hz, 1H, Ar-H), 8.27 (d, *J* = 8.1 Hz, 2H, Ar-H), 8.03 (d, *J* = 9.4 Hz, 2H, Ar-H), 7.93 (dd, *J* = 8.4 Hz, 2.6 Hz, 1H, Ar-H), 7.88 (d, *J* = 1.9 Hz, 1H, Ar-H), 7.83–7.78 (m, 1H, Ar-H), 7.73 (d, *J* = 8.3 Hz, 1H, Ar-H), 7.69–7.64 (m, 1H, Ar-H), 7.62 (t, *J* = 8.7 Hz, 1H, Ar-H), 3.79 (s, 2H, S-CH_2_). ^13^C NMR (150 MHz, DMSO-*d_6_*): *δ* 184.4, 159.0, 157.8, 155.8, 148.4, 148.1, 138.8, 138.2, 135.9, 133.9, 133.7, 132.8, 130.8, 130.5, 130.1, 127.2, 127.2, 126.5, 124.6, 124.6, 120.1, 119.6, 32.0.; HRMS (ESI) *m*/*z*: [M+H]^+^ calcd for C_23_H_17_Cl_2_N_8_O_4_S_2_,603.0187; Found, 603.0162.

#### 3.2.18. (.E)-N-(4-methyl-2-nitrophenyl)-2-(1-(2-nitrophenyl)-2-((5-(4-nitrophenyl)-4H-1,2,4-triazol-3-yl)thio)ethylidene)hydrazine-1-carbothioamide (**6r**)

Yield: 69%; ^1^H NMR (600 MHz, DMSO-*d_6_*): *δ* 13.33 (s, 1H, -NH), 11.06 (s, 1H, -NH), 10.72 (s, 1H, -NH), 8.77 (d, *J* = 7.1 Hz, 1H, Ar-H), 8.29 (d, *J* = 1.7 Hz, 1H, Ar-H), 8.24 (d, *J* = 9.6 Hz, 2H, Ar-H), 8.14 (dd, *J* = 7.8 Hz, 2.9 Hz 1H, Ar-H), 8.07 (dd, *J* = 7.5 Hz, 2.6 Hz 1H, Ar-H), 8.02 (d, *J* = 7.3 Hz, 2H, Ar-H), 7.99–7.93 (m, 1H, Ar-H), 7.91 (dd, *J* = 8.3 Hz, 1.5 Hz, 1H, Ar-H), 7.66 (dt, *J* = 8.5 Hz, 1.8 Hz 1H, Ar-H), 3.73 (s, 2H, S-CH_2_), 2.40 (s, 3H, CH_3_). ^13^C NMR (150 MHz, DMSO-*d_6_*): *δ* 184.6, 159.2, 158.0, 156.0, 148.3, 147.6, 139.0, 137.7, 135.8, 135.7, 135.3, 132.3, 132.2, 129.8, 127.4, 127.4, 126.8, 125.8, 125.7, 124.8, 124.8, 124.6, 31.0, 20.7; HRMS (ESI) *m*/*z*: [M+H]^+^ calcd for C_24_H_20_N_9_O_6_S_2_,594.0974; Found, 594.0960.

#### 3.2.19. (.E)-N-(4-nitrophenyl)-2-(1-(2-nitrophenyl)-2-((5-(4-nitrophenyl)-4H-1,2,4-triazol-3-yl)thio)ethylidene)hydrazine-1-carbothioamide (**6s**)

Yield: 63%; ^1^H NMR (600 MHz, DMSO-*d_6_*): *δ* 13.38 (s, 1H, -NH), 11.34 (s, 1H, -NH), 11.17 (s, 1H, -NH), 8.32 (d, *J* = 9.4 Hz, 2H, Ar-H), 8.19 (dd, *J* = 9.0 Hz, 2H, Ar-H), 8.12 (dd, *J* = 6.8 Hz, 1.9 Hz 1H, Ar-H), 8.08 (d, *J* = 9.6 Hz, 2H, Ar-H), 8.03 (dd, *J* = 8.5 Hz, 1.6 Hz 1H, Ar-H), 7.94–7.88 (m, 1H, Ar-H), 7.76 (d, *J* = 7.8 Hz, 2H, Ar-H), 7.64 (dt, *J* = 8.3 Hz, 1.7 Hz, 1H, Ar-H), 3.82 (s, 2H, S-CH_2_). ^13^C NMR (150 MHz, DMSO-*d_6_*): *δ* 184.4, 159.0, 157.8, 155.8, 148.1, 144.8, 144.1, 138.8, 135.6, 134.5, 132.1, 132.0, 127.2, 127.2, 126.6, 125.6, 125.0, 125.0, 124.6, 124.6, 124.4, 124.4, 31.0. HRMS (ESI) *m*/*z*: [M+H]^+^calcd for C_23_H_18_N_9_O_6_S_2_,580.0818; Found, 580.0806.

#### 3.2.20. (.E)-2-(1-(3-methoxyphenyl)-2-((5-(4-nitrophenyl)-4H-1,2,4-triazol-3-yl)thio)ethylidene)-N-(4-methyl-2-nitrophenyl)hydrazine-1-carbothioamide (**6t**)

Yield: 59%; ^1^H NMR (600 MHz, DMSO-*d_6_*): *δ* 13.37 (s, 1H, -NH), 11.14 (s, 1H, -NH), 10.81 (s, 1H, -NH), 8.85 (d, *J* = 7.7 Hz, 1H, Ar-H), 8.35 (d, *J* = 1.6 Hz, 1H, Ar-H), 8.30 (d, *J* = 9.4 Hz, 2H, Ar-H), 8.06 (d, *J* = 9.2 Hz, 2H, Ar-H), 7.97 (dd, *J* = 7.3 Hz, 1.8 Hz, 1H, Ar-H), 7.65 (dt, *J* = 7.7 Hz, 2.3 Hz, 1H, Ar-H), 7.43 (dd, *J* = 2.1 Hz, 1.3 Hz, 1H, Ar-H), 7.34 (dd, *J* = 6.3 Hz, *J* = 6.6 Hz, 1H, Ar-H), 7.10–6.99 (m, 1H, Ar-H), 3.74 (s, 3H, -OCH_3_), 3.80 (s, 2H, S-CH_2_), 2.45 (s, 3H, CH_3_). ^13^CNMR (150 MHz, DMSO-*d_6_*): *δ* 184.0, 160.5, 158.6, 157.4, 155.4, 147.7, 147.0, 138.4, 135.2, 135.1, 134.8, 129.6, 129.2, 126.8, 126.8, 125.1, 124.4, 124.2, 124.2, 120.3, 116.4, 113.1, 55.6, 31.6, 20.1. HRMS (ESI) *m*/*z*: [M+H]^+^ calcd for C_25_H_23_N_8_O_5_S_2_,579.1229; Found, 579.1213.

#### 3.2.21. (.E)-2-(1-(3-methoxyphenyl)-2-((5-(4-nitrophenyl)-4H-1,2,4-triazol-3-yl)thio)ethylidene)-N-(4-nitrophenyl)hydrazine-1-carbothioamide (**6u**)

Yield: 61%; ^1^H NMR (600 MHz, DMSO-*d_6_*): *δ* 13.40 (s, 1H, -NH), 11.34 (s, 1H, -NH), 11.07 (s, 1H, -NH), 7.76 (d, *J* = 6.9 Hz, 2H, Ar-H), 8.33 (d, *J* = 9.6 Hz, 2H, Ar-H), 8.13 (dd, *J* = 8.8 Hz, 2H, Ar-H), 8.08 (d, *J* = 9.2 Hz, 2H, Ar-H), 7.68 (dt, *J* = 8.7 Hz, *J* = 2.7 Hz, 1H, Ar-H), 7.44 (dd, *J* = 2.2 Hz, 1.4 Hz, 1H, Ar-H), 7.35 (dd, *J* = 6.8 Hz, 7.0 Hz, 1H, Ar-H), 7.13–6.95 (m, 1H, Ar-H), 3.81 (s, 3H, -OCH_3_), 3.78 (s, 2H, S-CH_2_). ^13^C NMR (150 MHz, DMSO-*d_6_*): *δ* 184.5, 161.0, 159.1, 157.9, 155.9, 148.2, 144.9, 144.2, 138.9, 135.3, 130.1, 127.3, 127.3, 125.1, 125.1, 124.7, 124.7, 124.5, 124.5, 120.8, 116.9, 113.6, 56.1, 32.1. HRMS (ESI) *m*/*z*: [M+H]^+^ calcd for C_24_H_21_N_8_O_5_S_2_,565.1073; Found, 565.1055.

### 3.3. Molecular Docking Protocol

Molecular docking was studied using MOE software to understand the binding mode of synthesized compounds against both the targeted enzymes in order to triangulate in vitro and in silico results well. The crystal structures of both targets were retrieved from the RCSB protein databank using the PDB codes 1ACL for AChE and 1P0P for BuChE. Using the default MOE-Dock module parameters, the crystallographic structures and all synthesized compounds were protonated and energy was minimized, resulting in optimized enzyme and compound structures. After that, the optimized enzyme and compound structures were used in a docking study. Comprehensive details of the docking protocol are included in our previous investigations [33,34].

### 3.4. Acetylcholinesterase Activity Assay Protocol

Based on previously described methods, in vitro study of AChE inhibitory profile was calculated [35,36]. In order to make a stock solution, compounds being analyzed were dissolved in DMSO (1 mg/mL). In addition, the working solutions were also prepared by using serial dilution (1–100 μg/mL). The solution of AChE enzyme (20 μL; 0.1 U/mL), analogues being tested with different concentration and buffer of sodium phosphate (150 μL; pH 8.0; 0.1 M) were preincubated appropriately at 25 °C. The process was initiated as DTNB (10 mM; 10 μL) and AChEI (14 mM; 10 μL) were accumulated. The resulting residue was mixed (by using cyclomixer) and put on incubation for 10 min at 25 °C. Instead of compounds being tested using 10 μL DMSO, the absorbance against blank reading was calculated with a microplate reader at 410 nm. Using the given formula as given, inhibition and IC_50_ values were measured in comparison to donepezil (0.01–100 μg/mL) as the reference standard (Equation (1)).
(1)$$\%\mathrm{Inhibition}=\mathrm{Absorbance}\;\mathrm{of}\;\mathrm{control}-\mathrm{Absorbance}\;\mathrm{of}\;\mathrm{compound}\times \frac{100}{\mathrm{Acontrol}}$$


By plotting a nonlinear graph between inhibition and concentration, IC_50_ was calculated using GraphPad Prism 5.3.

### 3.5. Butyrylcholinesterase Activity Assay Protocol

In order to explore the inhibition profile of in vitro BuChE enzyme, a similar procedure was adopted. For measurement of BuChE activity, the solution containing BuChE enzyme was used [35,36].

## 4. Conclusions

Triazole-based thiosemicarbazone derivatives (**6a–u**) were synthesized and screened for potential to inhibit activities of acetylcholinesterase and butyrylcholinesterase enzymes. All the synthetic derivatives (except compounds **6c** and **6d**, which were found to be completely inactive) displayed moderate to good inhibitory activities having an IC_50_ values ranging from 0.20 ± 0.050 to 12.20 ± 0.30 µM (against AChE) and 0.40 ± 0.050 to 14.10 ± 0.40 µM (against BuChE) compared to the standard drug donepezil (IC_50_ = 2.16 ± 0.12 (AChE) and 4.5 ± 0.11 µM (BuChE)). Among the series, derivative **6q** (IC_50_ = 0.20 ± 0.050 µM) was the most potent inhibitor of acetylcholinesterase enzyme, while derivative **6o** (IC_50_ = 0.40 ± 0.050 µM) was the most active inhibitor of butyrylcholinesterase enzyme. The binding interactions of most active compounds with the active site of enzymes were established with the help of molecular docking studies.

## Data Availability

Not applicable.

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
