# Peer review of "Synthesis of New Triazole-Based Thiosemicarbazone Derivatives as Anti-Alzheimer’s Disease Candidates: Evidence-Based In Vitro Study"

_molecules, 2022, doi:10.3390/molecules28010021_

Round 1
Reviewer 1 Report (New Reviewer)
Figure 6 can be improved cause is blurry. Some titles in the spectra description must be improved too, row materials commercial sources must be included, and chemical structure are in a different format. It´s an interesting job.

Author Response
Comments and Suggestions for Authors
Figure 6 can be improved cause is blurry. Some titles in the spectra description must be improved too, row materials commercial sources must be included and chemical structure are in a different format. It´s an interesting job.
Reply: The figure-6 was revised. Titles in spectral description were also improved. Sources of row material incorporated and the structure is now in one format and corrected according to the kind reviewer suggestions.
Reviewer 2 Report (New Reviewer)
References 29 and 30 are not suitable. for example you can mention the references which used thiosemicarbazone in their works like
https://doi.org/10.1080/10406638.2021.2014537
https://doi.org/10.1002/jhet.4548
Scheme 1 mentioned capital letters A and B in benzene ring did you explained about this in the manuscript
Author Response
Comments and Suggestions for Authors
References 29 and 30 are not suitable. For example you can mention the references which used thiosemicarbazone in their works like
https://doi.org/10.1080/10406638.2021.2014537
https://doi.org/10.1002/jhet.4548
Reply: The references 29 & 30 were revised by incorporating new references which used thiosemicarbazone as per kind suggestion
Scheme 1 mentioned capital letters A and B in benzene ring did you explained about this in the manuscript.
Reply: Now corrected according to the kind reviewer suggestion.
Reviewer 3 Report (New Reviewer)

Author Response
In this paper the synthesis of novel triazole-based thiosemicarbazone derivatives is described
in this study, as well as their characterization using spectroscopic methods like 1HNMR,
13CNMR, and HRMS (ESI). Images of the spectra were not displayed by the authors.
Reply: The spectral data of representative compounds were provided in supplementary information as per kind suggestion.
In addition, the results are very briefly discussed without significant explanation of the
influence of structure on inhibitory activity.
Reply: The Structure activity relationship (SAR) studies were carried out for all synthesized analogues in which we mainly discuss the influence of structures of the synthesized analogue on inhibitory potentials of targeted AChE & BuChE enzymes.
An in silico study was also done only for three compounds, while the others are missing. Why? The discussion is also very short and without significant explanation.
Reply: The in silico study was only carried out for the most potent molecules therefore only three compounds were done. Now the discussion part is further explained and the study of said three compounds against butyrylcholinesterase enzymes was also incorporated as per kind reviewer suggestion.
Round 2
Reviewer 3 Report (New Reviewer)
In my opinion, the paper can be accepted for publication in the journal Molecules.
This manuscript is a resubmission of an earlier submission. The following is a list of the peer review reports and author responses from that submission.
Round 1
Reviewer 1 Report
The submission provides information about a series of triazole-based thiosemicarbazone derivatives (1-21) and their in vitro inhibitory efficacy against the AChE and BuChE enzymes. The SAR shows that all of the other synthetic derivatives, except for compounds 3 and 4, had moderate to good inhibitory activity. The authors additionally performed molecular docking studies to further understand the structure-activity relationship. The article is well-written, with clear explanations of the findings and adequate graphical representations
Despite the benefits of this work, the authors need to address a few key points:
1/ Standard drug Donepezil has an IC50 value of 0.016 ± 0.12 μM. Have you tested your compounds on a 0.01 μM level?
2/ Some recent references should be added in the introduction and docking part: DOI: https://doi.org/10.1016/j.jksus.2021.101632; DOI: https://doi.org/10.1016/j.molstruc.2022.133845.
3/ Why do you select to dock just compounds (14), (15), and (17), and what about the affinity values?
4/ 2.1. Chemistry is just repeating the synthetic procedure as 3.2. The general procedure of 1,2,4-triazole bearing thiosemicarbazone derivatives (1-21).
5/ In section 2.1 (Chemistry), perhaps discuss the chemistry of molecules.
Author Response
Reviewer-1:
Comments and Suggestions for Authors
The submission provides information about a series of triazole-based thiosemicarbazone derivatives (1-21) and their in vitro inhibitory efficacy against the AChE and BuChE enzymes. The SAR shows that all of the other synthetic derivatives, except for compounds 3 and 4, had moderate to good inhibitory activity. The authors additionally performed molecular docking studies to further understand the structure-activity relationship. The article is well-written, with clear explanations of the findings and adequate graphical representations
Despite the benefits of this work, the authors need to address a few key points:
- Standard drug Donepezil has an IC50value of 0.016 ± 0.12 μM. Have you tested your compounds on a 0.01 μM level?
Reply: Yes, we have checked the activity of our compound in comparison to the standard drug Donepezil.
- Some recent references should be added in the introduction and docking part: DOI:https://doi.org/10.1016/j.jksus.2021.101632; DOI: https://doi.org/10.1016/j.molstruc.2022.133845.
Reply: These references were incorporated in docking part as per kind suggestion
- Why do you select to dock just compounds (14), (15), and (17), and what about the affinity values?
Reply: Thank you for your insightful comments. Based on the ranking of the IC50 value, the compounds for molecular docking have been chosen.
- 2.1. Chemistry is just repeating the synthetic procedure as 3.2. The general procedure of 1,2,4-triazole bearing thiosemicarbazone derivatives (1-21).
Reply: The synthetics procedure as 3.2. The general procedure of 1,2,4-triazole bearing thiosemicarbazone derivatives was modified to remove repetition as per kind suggestion
- In section 2.1 (Chemistry), perhaps discuss the chemistry of molecules.
Reply: In section 2.1 (chemistry) the chemistry of molecules was incorporated as per kind suggestion
Reviewer 2 Report
In my opinion, the choice of research topic is justified because both triazoles and thiosemicarbazones exhibit biological activity, including action as inhibitors of cholinesterases. The combination of both blocks in one molecule may result in increased hybrid action, as the authors intended. An additional advantage is the lack of literature reports on this subject. The clear presentation of the results deserves recognition. It is worse with their interpretation. The finding that the activity of the compounds depends on the ring substituents is insufficient. The publication lacks the results of calculations (eg QSAR) that would facilitate the selection of appropriate test objects and the interpretation of the results. Although a docking simulation was performed, the conclusions are not clear and did not lead to the synthesis of new, more active compounds. Therefore, in my opinion, the work can be published, but requires thorough changes in the subject of interpretation of the results. Other comments are provided below.
Line 50 „These enzymes…” but what enzymes?
Line 82 It does not seem necessary to present here the achievements of the research group. Instead, it will be better to present missing at work results for holinoesterases. Therefore it would be better to make the following changes: „…heterocyclic compounds as potent inhibitors of cholinesterases [relevant ref].
Line 84 It should be: „…thiosemicarbazone compounds (Figure-3) [insert missing ref]
Figure 3. „Previously reported analogues” – It is unclear if these are just your results or the best results known in the literature. This should be clarified and appropriate references should be made (see the note above line 84). Check the correctness of the given structures. The thiosemicarbazide structure should be corrected (S atom, R group).
Line 129. „Higher potency 128 of these two derivatives may be due to the presence of electron withdrawing groups.” – explanation insufficient, other molecules also have EWG.
Line 130 „…derivative 9 (IC50 = 12.20 ± 0.30 & 14.10 ± 0.40 µM respectively) having bromo 130 group at para-position of ring B and nitro group at para-position of ring C..” – the description of compound 9 is incorrect.
Line 137 Should be orto- position of bromine atom, not para.
Line 144 „The difference in the activity of these three derivatives may be due to the different position and nature of the substituent/s on phenyl ring B” – Conclusions are too general, the interpretation should be enriched. The same remark applies to the similar conclusions that are often repeated in the text.
Line 187 Which compounds have been optimized? Please explain.
Lines 191-196 „Generally, we have noticed that all the compounds ….inhibitory potential. Mostly, the substituted group belongs to withdrawing or donating groups.” – This part of the text is not clear.
Line 210-212 Fragment „The other compounds in the series also showed similar potential against…. against the AChE and (Figure-10D) against the BuChE.” This part of the text is not clear.
Line 212-214 The final conclusion is illogical given the earlier statement: „even though this compound lack of the substituted groups at their 3rd end, but still showed the best inhibitory potential” (lines 203-204). Moreover, compound 9, as the weakest inhibitor, has the most substituents on the B and C phenyl rings.
Line 226 The „general procedure” can not be a repetition of the earlier text (lines 91-101). Basic information about the procedure is not provided, i.e. the amount of reagents, solvents, proportions (moles), processing and purification methods, yields of each step. This should be completed. It is assumed that all previously undescribed compounds should be characterized. Were the new intermediates isolated and characterized by NMR? If so, why there is no description of the spectra.
In the part „Spectral Analysis” explain which steps are related to the given yields.
Why in all calculated and found HRMS (EI) peak masses were obtained as [M + H] instead of [M]?
Author Response
Reviewer-2:
Comments and Suggestions for Authors
In my opinion, the choice of research topic is justified because both triazoles and thiosemicarbazones exhibit biological activity, including action as inhibitors of cholinesterases. The combination of both blocks in one molecule may result in increased hybrid action, as the authors intended. An additional advantage is the lack of literature reports on this subject. The clear presentation of the results deserves recognition. It is worse with their interpretation. The finding that the activity of the compounds depends on the ring substituents is insufficient. The publication lacks the results of calculations (eg QSAR) that would facilitate the selection of appropriate test objects and the interpretation of the results. Although a docking simulation was performed, the conclusions are not clear and did not lead to the synthesis of new, more active compounds. Therefore, in my opinion, the work can be published, but requires thorough changes in the subject of interpretation of the results. Other comments are provided below.
Line 50 „These enzymes…” but what enzymes?
Reply: The cholinesterase enzymes such as AChE and BuChE
Line 82 It does not seem necessary to present here the achievements of the research group. Instead, it will be better to present missing at work results for cholinesterases. Therefore it would be better to make the following changes: „…heterocyclic compounds as potent inhibitors of cholinesterases [relevant ref].
Reply: Heterocyclic compounds as potent inhibitors of cholinesterase [24-28] were incorporated as per kind suggestion
Line 84 It should be: „…thiosemicarbazone compounds (Figure-3) [insert missing ref]
Reply: Line 84, C=S is mistakenly written as C=O, therefore it is corrected and also relevant references related to thiosemicarbazone and triazole were incorporated as per kind suggestion
Figure 3. "Previously reported analogues” – It is unclear if these are just your results or the best results known in the literature. This should be clarified and appropriate references should be made (see the note above line 84). Check the correctness of the given structures. The thiosemicarbazide structure should be corrected (S atom, R group).
Reply: Appropriate references were made and also the thiosemicarbazide structure was corrected as per kind suggestion
Line 129. „Higher potency 128 of these two derivatives may be due to the presence of electron withdrawing groups.” – explanation insufficient, other molecules also have EWG.
Reply: Sufficient explanation related to higher potency of 17 and 15 derivatives bearing EWD groups was incorporated as per kind suggestion
Line 130 „…derivative 9 (IC50 = 12.20 ± 0.30 & 14.10 ± 0.40 µM respectively) having bromo 130 group at para-position of ring B and nitro group at para-position of ring C..” – the description of compound 9 is incorrect.
Reply: The description related to compound 9 was revised as per kind suggestion
Line 137 Should be orto- position of bromine atom, not para.
Reply: Line 137 describing the description of derivative 1 having bromo-substitution at ortho-position of ring B which was mistakenly written as para-bromo moiety. The sentence with revised as per kind suggestion
Line 144 „The difference in the activity of these three derivatives may be due to the different position and nature of the substituent/s on phenyl ring B” – Conclusions are too general, the interpretation should be enriched. The same remark applies to the similar conclusions that are often repeated in the text.
Reply: The sentence in Line 144 describing the difference of activity among three derivatives was revised as per kind suggestion
Line 187 which compounds have been optimized? Please explain.
Reply: Authors appreciate the comment. Authors have optimized all of the triazole based thiosemicarbazone derivatives via MOE-Potential Energy Minimize module, and then docked based on the co-crystal part, and then used only the top 2 in each series for molecular docking insight.
Lines 191-196 „Generally, we have noticed that all the compounds ….inhibitory potential. Mostly, the substituted group belongs to withdrawing or donating groups.” – This part of the text is not clear.
Reply: Authors have revised the above mentioned sentence, please see the revision in line 249-253.
Line 210-212 Fragment „The other compounds in the series also showed similar potential against…. against the AChE and (Figure-10D) against the BuChE.” This part of the text is not clear.
Reply: Authors have revised the abovementioned sentence, please see the revision in line-268-271.
Line 203-205 The final conclusion is illogical given the earlier statement: „even though this compound lack of the substituted groups at their 3rd end, but still showed the best inhibitory potential” (lines 203-204). Moreover, compound 9, as the weakest inhibitor, has the most substituents on the B and C phenyl rings.
Reply: Thank you for your valuable comment. Authors strongly agree with your critical analysis. Basically, the authors assume that might be it will work in this way. Authors have stated in the main text, what the authors assume during docking study and present the mechanism in might be manner. Sometime, it more than hard to correlate well the in-silico and experimental results.
Line 226 The “general procedure” cannot be a repetition of the earlier text (lines 91-101). Basic information about the procedure is not provided, i.e. the amount of reagents, solvents, proportions (moles), processing and purification methods, yields of each step. This should be completed. It is assumed that all previously described compounds should be characterized. Were the new intermediates isolated and characterized by NMR? If so, why there is no description of the spectra.
Reply: The “general procedure” was modified to remove repetition as per kind suggestion. In addition, basic information about procedure including amount of reagents, solvents, proportions, processing and purification methods and yields of each step was incorporated.
In the part “Spectral Analysis” explain which steps are related to the given yields.
Reply: In the final step, triazole-based schiff base intermediate reacts with different substituted phenyl isothiocyanates in THF along with few drops of triethylamine to obtain triazole-based thiosemicarbazone in given yield as mentioned in spectral analysis for each synthesized compound.
Why in all calculated and found HRMS (EI) peak masses were obtained as [M + H] instead of [M]?
Reply: HRMS (ESI) spectrometry was carried out for synthetics compounds in which the ions observed in all calculated and found may be quasi molecular ion created by the addition of hydrogen cation and was denoted as [M+H]+. It could be shown that probably an intermolecular hydrogen cation transfer to synthetics compound from solvent to give rise quasi molecular ions.
Reviewer 3 Report
Comments: Ullah, Rahim et al. describes the ChE activity of some new triazole based thiosemicarbazone derivatives. Molecular docking and in vitro assays were demonstrated.
The IC50 results were promising concerning AChE and BChE inhibitory activity, using the well-known drug Donepezil as positive control.
I think this work, as it is, cannot be published in Molecules magazine. The following issues addresses my decision:
The manuscript is poorly written, with serious grammar mistakes; sentences without any sense; English should be improved significantly;
Examples: Introduction: the first sentence in not clear; Sentence in line 50: what are you talking about?
Regarding acronyms: if you decided to use them use it, please be coherent (use always AChE for instance and not acetylcholinesterase in one sentence and in another AChE);
Aß: what is that? Attention with the acronyms;
Line 65: several drugs for AD? I think this is not correct (there are only the ones that you described);
Subtitle of Figure 1: drugs (not drug!)
Line 71: remove of;
Line 84: not clear;
Figure 3: is the first scaffold, right? C=O or C=S?; second scaffold: why BuChE activity is in percentage micromolar?
Results and Discussion: In my opinion structures (I to V) should be numbered (15); in text when you refer to a structure, please add the scheme where we can find them;
3hrs: ? (3 hours)
Line 99: remove reacted and
Scheme 1: R is always the same in all the intermediates and final products, right? Remove R and put the nitro aryl group; You can discriminate the groups of R1 and R2 and the corresponding yields of the compounds; the yields should be referred in the paper (not only in experimental section);
Table 1: S.NO? and N.A.?
AChE from what? Electric eel? Human? The same for BuChE;
Point 2.2: when you refer the activity of the derivative you should refer for witch enzyme and where the reader can see the result (table 1, for instance);
Figure 4: SAR analysis incomplete! Put in the figure the most potent derivatives in order for each ring (as it is, completely useless);
Point 2.2.1.1: major problem of the discussion: tedious description of the compounds and unworkable; put all that written information in Figure 4 in the mode advised before; You refer compound 17 as the best one regarding AChE activity; and compounds 13, 14, 15 and 16?
Every time that a compound is described please put reference to where the reader can find it (table1, for instance);
Line 136 starts: If we comparing derivative… and ends with ring C???? No sense…
Figures 5, 6, 7, 8 and 9: remove and put the SAR study complete in Figure 4;
Figure 6: you should compare (1) with (8) (8 and 12 are similar);
Figure 7: no sense: the activity is similar; (the same for Figure 8 with 13, 18 and
20);
Line 181: effect greatly affect?
Docking study: line 187: optimized compounds? How so?
Sentence in line 191: no sense;
You refer before rings A, B and C to differentiate the derivatives; why now refer 3rd end etc??
Line 210: no sense;
Figure 10: regarding AChE why dock compound 14 and not the 17 witch is the best in vitro?
3. In extraction and isolation section CC should be identified (chromatographic column?);
4. in the same section please rephrase “… developing solvent…”; maybe replace by “mobile phase” or just “solvent”;
5. Concerning section 3.2. ChE activities, according to Table, the IC50 value of compound (2) for BChE activity is 167.3 !mol/L and not 46.2 as you describe in the text; please clarify this crucial issue;
6. In the same section, page 18, line 10, (Figure 2) should be add in the end of this paragraph and not in the paragraph above; please check;
7. Page 20, line 14, (Table 3) should be add in the end of the paragraph; please check;
8. Page 22, line 4, this sentence should be rephrased; IC50 values concerning
AchE? BChE? please check;
9. Page 23, line 11, please check; I think the correct value for (2) (according to Table 2) is 107.3 and not 46.2; and please rephrase the sentence, since this value is not so notable at all;
Author Response
Reviewer-3:
Comments and Suggestions for Authors
Comments: Ullah, Rahim et al. describes the ChE activity of some new triazole based thiosemicarbazone derivatives. Molecular docking and in vitro assays were demonstrated.
The IC50 results were promising concerning AChE and BChE inhibitory activity, using the well-known drug Donepezil as positive control. I think this work, as it is, cannot be published in Molecules magazine. The following issues address my decision:
The manuscript is poorly written, with serious grammar mistakes; sentences without any sense; English should be improved significantly;
Reply: The sentences without sense, serious grammar mistakes were corrected as per kind suggestion
Examples: Introduction: the first sentence in not clear; Sentence in line 50: what are you talking about?
Reply: The sentence in line 50 was modified as per kind suggestion
Regarding acronyms: if you decided to use them use it, please be coherent (use always AChE for instance and not acetylcholinesterase in one sentence and in another AChE);
Reply: The kind suggestion was incorporated by using AChE throughout manuscript uniformly
Aß: what is that? Attention with the acronyms;
Reply: Aß stand for beta amyloid protein
Line 65: several drugs for AD? I think this is not correct (there are only the ones that you described);
Reply: These are already reported drugs of alzheimer disease approved by FDA. Each drugs has its own side effect, therefore we have focus on triazole-based thiosemicarbazone drug with no side effect.
Subtitle of Figure 1: drugs (not drug!)
Reply: The word “drug” was corrected as “drugs” as per kind suggestion
Line 71: remove of;
Reply: The sentence in line 71 was removed as per kind suggestion
Line 84: not clear;
Reply: sentence in line 84 was revised to make sense as per kind suggestion
Figure 3: is the first scaffold, right? C=O or C=S?; second scaffold: why BuChE activity is in percentage micromolar?
Reply: In figure 3 the first scaffold is mistakenly written as C=O which is corrected as C=S. furthermore, BuChE activity of related scaffold was also revised
Results and Discussion: In my opinion structures (I to V) should be numbered (1-5); in text when you refer to a structure, please add the scheme where we can find them;
Reply: The structure in result and discussion (I to V) were numbered as 1-5 in the text and then synthetic compounds were numbered as 6a-u as per kind suggestion
3hrs: ? (3 hours)
Reply: The 3hrs was revised with 3 hours as per kind suggestion
Line 99: remove reacted and
Reply: Line 99: the “reacted and” was removed as per kind suggestion
Scheme 1: R is always the same in all the intermediates and final products, right? Remove R and put the nitro aryl group; You can discriminate the groups of R1 and R2 and the corresponding yields of the compounds; the yields should be referred in the paper (not only in experimental section);
Reply: The R (4-nitro aryl ring) is incorporated in the scheme as per kind suggestion
Table 1: S.NO? and N.A.?
Reply: In table 1, S.NO shows the numbers of synthetics compounds (1-19) (S. NO changed with compounds) and N.A stands for “not active” compound which don’t show potency
AChE from what? Electric eel? Human? The same for BuChE;
Reply: The suggestion was incorporated in general information under experimental section
Point 2.2: when you refer the activity of the derivative you should refer for witch enzyme and where the reader can see the result (table 1, for instance);
Reply: The suggestion was incorporated
Figure 4: SAR analysis incomplete! Put in the figure the most potent derivatives in order for each ring (as it is, completely useless);
Reply: SAR analysis of figure 4 was incorporated as per kind suggestion
Point 2.2.1.1: major problem of the discussion: tedious description of the compounds and unworkable; put all that written information in Figure 4 in the mode advised before; You refer compound 17 as the best one regarding AChE activity; and compounds 13, 14, 15 and 16?
Reply: SAR studies for all synthetics compounds was put in figure 4 as per kind suggestion
Every time that a compound is described please put reference to where the reader can find it (table1, for instance);
Reply: The reference (table-1) was added when the compound is described as per kind suggestion
Line 136 starts: If we comparing derivative… and ends with ring C???? No sense…
Reply: Line 136 was revised to make some sense as per kind suggestion
Figures 5, 6, 7, 8 and 9: remove and put the SAR study complete in Figure 4;
Reply: The figures 5, 6, 7, 8 and 9 were removed and SAR study was incorporated in figure 4 as per kind suggestion
Figure 6: you should compare (1) with (8) (8 and 12 are similar);
Reply: Figure 6 was deleted and SAR study was incorporated in figure 4. Moreover, derivative 1 was compared with derivative 8 as per kind suggestion
Figure 7: no sense: the activity is similar; (the same for Figure 8 with 13, 18 and
20);
Reply: Figure 7 and figure 8 were deleted as per suggestion of reviewer. Furthermore, sentences related to figures 7 and 8 were also revised to make sense as per kind suggestion
Line 181: effect greatly affect?
Reply: Line 181 was revised to make sense as per kind suggestion
Docking study: line 187: optimized compounds? How so?
Reply: Thank you for your comment. Authors have optimized all of the triazole based thiosemicarbazone derivatives via MOE-Potential Energy Minimize module, and then docked based on the co-crystal part, and then used only the top 2 in each series for molecular docking insight.
Sentence in line 191: no sense;
Reply: Author state straight apology for misstatement. Authors have revised the following sentence accordingly.
You refer before rings A, B and C to differentiate the derivatives; why now refer 3rd end etc??
Reply: good suggestion we will try next time in future we will use 1st, 2nd and 3rd ends for differentiation of derivatives
Line 210: no sense;
Reply: Line 210 was revised to make some sense as per kind suggestion
Figure 10: regarding AChE why dock compound 14 and not the 17 witch is the best in vitro?
Reply: Authors highly appreciate and agree with the review comment aroused here. Author docked compound 14 and not compound 17 just for comparative study. To show the impact of changing the position of the substituted group, we didn’t dock compound 17. Because it already included in the discussion against the BChE.
In extraction and isolation section CC should be identified (chromatographic column?);
Reply: The identification of column chromatography in extraction and isolation was made as per kind suggestion
In the same section please rephrase “… developing solvent…”; maybe replace by “mobile phase” or just “solvent”;
Reply: The solvents which were used in extraction process and column chromatography were added as per kind suggestion.
Concerning section 3.2. ChE activities, according to Table, the IC50 value of compound (2) for BChE activity is 167.3 !mol/L and not 46.2 as you describe in the text; please clarify this crucial issue;
Reply: The IC50 of compound is corrected in the text and table.
In the same section, page 18, line 10, (Figure 2) should be add in the end of this paragraph and not in the paragraph above; please check;
Reply: Corrected according to the kind reviewer suggestion.
Page 20, line 14, (Table 3) should be added in the end of the paragraph; please check;
Reply: I donot understand, there is no table-3 in my manuscript.
Page 22, line 4, this sentence should be rephrased; IC50 values concerning AchE? BChE? please check;
Reply: The sentence concerning AChE and BuChE was re-phrase as per kind suggestion
Page 23, line 11, please check; I think the correct value for (2) (according to Table 2) is 107.3 and not 46.2; and please rephrase the sentence, since this value is not so notable at all;
Reply: The IC50 of compound is corrected in the text and table.
Round 2
Reviewer 2 Report
Dear Authors,
please refer to the new manuscript notes.
26-40 The numbering of the compounds should be changed.
27 should be: „…derivatives 6 which was characterized…”
51 The abbreviations should be explained.
Figure 3. According to the reference [30], the drawn thiosemicarbazide structure should contain a chlorine atom in the quinoline ring. The given range of values 1.33-32.6 does not refer to the IC50 but to Inhibition (%).
91-109 This detailed section should be included in the experimental part as „general procedure”, not in ‘Results and discussion”. Were intermediates isolated and purified? If so, the authors still do not provide the yields of stages and methods of purifying intermediates. If not, how were side products (triethylamine chlorides or bromides, water) or excess triethylamine, acetic acid removed? It should be described. If the obtained intermediates and procedures are known and described in the literature, you should add only ref. The authors should seriously correct the given values, e.g. the 0.5 mmol of 1 can not give 0.8 mmol of 2! or 0.5 mmol of 3 can not give 0.7 mmol of 4 !! etc. etc. The chemical part requires thorough improvement and completion.
129 „As for compound 6n” – to remove
114-144 What is the purpose of this new detailed description of the spectra of compound 6n?
159 The abbreviation SAR should be explained.
160 „Synthetics compounds” or „general structure of obtained compounds”?
161 „..entire parts of synthetic compounds..”?
164 „..constant and variation..” or „…constant the variation…”?
192 „…groups at of ring C…”?
Figure 4 It should be: „..introduction of EWG groups such as Cl, NO2, at various…”
197 „ortho-nitro substitutions on ring C”?
215 „electro with drawing” ?
238 It should be: „…nature of substituents considerably influence the inhibitory…”
248-253 It should be: „In general, we found that all of the compounds in the series, having different substituted groups at all three ends of the compound (according to the scheme), where one end has a nitro group, the second end has a halogen group, and the most important end (third) has a different substituted group, have a different inhibitory potential against the target. The groups typically belong to electro withdrawing or electro donating groups.”
262 „…activating and deactivating groups at their 1st and 2nd end…”? There are only deactivating groups: NO2 and Cl.
269 „as shown in as shown in”?
270-271 „The computational and bioassay studies revealed that compounds with a strong substituted group had the best potential against both targets enzymes.” Are the compounds 6o and 6i an exceptions? It is about „strong substituted group” or „strong substituted groups”, which phenyl group? This sentence are not supported by the results.
286-299 The description is too general (see the note above for lines 91-109).